# Context and Minimalism: User Evaluations on Two Interpersonal Telepresence Systems

Category: Research

## ABSTRACT

We present a field study with 10 pairs of adults to employ two prototypes of an interpersonal telepresence system: one catered to the Viewer, the individual experiencing a remote environment, and one catered to the Streamer, the individual sharing the remote experience to the Viewer through technological means. Based on previous work, our design choices reflect common values found from both perspectives of the Viewer and Streamer. We then seek to identify the key value tensions and trade-offs in the designs through the employment of similar design choices. We then demonstrate how, through an applied scenario, we learned that users converge on environmental context and minimalism being the prime factors that should influence a general framework for what considerations need to be held when designing a telepresence system catered to one-to-one interactions.

**Keywords:** Telepresence, Live-streaming, Remote experiences

**Index Terms:** Human-centered computing—Visualization—Visualization techniques;

## 1 INTRODUCTION

Physical presence has become a point of contention in multiple communities due to the COVID-19 Pandemic. Many people who were previously present with others in the physical world are now either confined to their living space or are concerned about the health effects that re-involving themselves in society may yield [1]. To combat this confinement, many are limited to virtual forms of communication, such as video-conferencing applications like *Zoom*, and interactions. To mitigate the lack of in-person interactions, the need for remote collaboration and interaction techniques is becoming increasingly important. Due to this need, we are witnessing an emergence of remote interaction technologies ranging from extending video conferencing capabilities to mixed reality collaboration experiences.

The facilitation of developing methods to simulate presence in a remote environment has led to the ideation of telepresence, a term coined by Marvin Minsky in 1980 to denote the use of technology to simulate a remote physical location despite not being physically located there [11]. This provided the formal definition for researchers to begin conceptualizing and applying their techniques to further the research of telepresence-related tasks [9, 14]. The current state of telepresence is now spanning a multitude of fields and disciplines such as medical and industrial applications. Within each field, we are witnessing a range of novel implementations that collectively propose effective remote experiences. However, when considering interpersonal telepresence systems, telepresence systems that hone on one-to-one remote interactions in which one individual attempts to experience a remote environment with an individual in the remote environment, we are met with concerns regarding how socially present does an remote individual feel and how socially comfortable is the physically present individual when hosting the technology required to fulfill the remote experience [4–6, 8, 12–16, 18].

With novelty having been a main focal point within the telepresence research community, we are now witnessing a divergence in what aspects of these telepresence systems need to be considered. The first is, are researchers considering the social effects of participants being involved in these telepresence systems, and, by ex-

tension, interpersonal telepresence systems? The second aspect calls researchers to consider if the technological implementations being employed are what end-users truly desire and value in a telepresence system [14].

To answer both these questions, our research intends to investigate the social and technological variables to consider in the development cycle of a future interpersonal telepresence system. To achieve this, we carried out a study with two interpersonal telepresence systems, each designed to be used by a pair of participants. The purpose for designing two interpersonal telepresence systems is to provide two platforms that cater to the two types of users of our systems: the Streamer and the Viewer. We denote the Streamer as the individual physically present in the environment to administer the technology necessary to fulfill the remote experience between them and the Viewer. We denote the Viewer as the individual utilizing technology to view and possibly interact with the remote environment.

Both systems we propose were designed with the idea of minimalism in terms of hardware and equipment needed to carry out the remote experience. We denote the two systems as either being Viewer-centric or Streamer-centric. Each design is based on techniques employed in previous literature. Our end goal was to expose our participants to two (2) variations of interpersonal telepresence setups and then prompt them on their experience as well as any improvements they would prefer in an effort to draw conclusions towards an optimal system. We propose the following research questions to be answered through our study:

- RQ1: How do Streamer participants approach social interactions while using the interpersonal telepresence systems?

- RQ2: In what real-world contexts do Streamers and Viewers truly value utilizing an interpersonal telepresence system?

- RQ3: How can we balance social presence of the Viewer and social comfort of the Streamer in a telepresence system?

As a result of our study, we found that users converged towards the idea that interpersonal telepresence setups should be context-sensitive. Furthermore, in regard to the interactions and technologies utilized, interpersonal telepresence setups should allow the Streamer and Viewer to maintain a level of interactive independence from one another. We discuss in future sections the finer details of our results as well as the implications for future interpersonal telepresence systems.

## 2 RELATED WORK

Previous work on interpersonal telepresence systems show that the Viewer's needs are prioritized [14]. With this in mind, we sought to develop and deploy two systems catering to each stakeholder separately. In this section, we review the relevant literature in the areas of telepresence and computing devices.

### 2.1 Identifying The Appropriate Telepresence Setup

Telepresence refers to the ability for someone to feel as if they are present in another physical space without actually being there. It can be supported in many forms, from the use of 3D avatars in virtual reality [10] to conventional video-conferencing solutions [6] to full 360-degree video streaming [18]. Our study uses a form

of video conferencing since it has been found to socially connect participants more than virtual reality along with providing a higher fidelity system that is more ubiquitous and familiar to participants, along with its ethical and social risks being more known than virtual reality telepresence [2]. As found by Tang et al. and Heshmat et al., 360-degree video-streaming provides more social connection between the Viewer and Streamer due to more visual immersion at the expense of verbal communication, which would require additional hardware to support such a form of communication [4, 18]. In contrast to these designs, Teo et al. conducted a collaboration study that compares the use of 360-degree systems to 3D scene reconstruction systems for collaboration [20]. With these systems employing nonverbal communication techniques (i.e. hand gestures and visual cues), social presence and task completion rate was found significantly higher for the 360-degree camera setup.

Communication is another prime element to these designs, with video conferencing delivering audio communication. Many forms of video conferencing exist, ranging from a regular video call using proprietary applications [16] or through off-the-shelf software such as *Skype* [1] [8, 17, 19].

In the past, interpersonal telepresence scenarios centralized on a certain task or series of tasks [5, 6, 8, 16, 18]. We are interested in observing the experiences that users undergo while applying such setups through a real-world experience and learn of what technologies and scenarios users truly value. Considering this, we employed two setups; one using both video conferencing and a 360 camera to cater to the Viewer since it provides more Viewer immersiveness, and the other using only video conferencing to cater to the Streamer as it requires less items for the Streamer to hold/carry.

## 2.2 Choosing Necessary Hardware

The key piece of hardware used to provide the video feed in many systems is a camera. However, in various telepresence setups, at least one user of the system is mobile (like in our system), leading to the use of cameras such as 360-degree cameras [7, 8, 18], smartphone cameras [6], or even hand-held cameras [5]. Camera quality is important to consider when using mobile cameras, as Kim et al. [8] shows that when presented with various media types, the Viewer chose other media types over video feed since it was of notably poor quality, indicating that good camera quality would be vital to a system.

To facilitate communication, smartphones [6, 8, 16, 18], desktops [8, 15], and even telepresence robots [4, 5] have been the main media that previous designs have centered on. Smartphones and desktops provide varying functionality that support communication efficiently (i.e. internet connection and an integrated camera), with desktops providing larger displays with little mobility and smartphones vice versa, but telepresence robots more so serve as a medium of physically supporting and moving hardware.

With all these designs in mind, the hardware in both of our telepresence setups are similar; they both share a desktop for the Viewer and a smartphone for the Streamer; the difference being that the Viewer centric setup also has an additional smartphone and 360-degree camera to facilitate immersiveness.

## 3 METHODS

Previous work in the interpersonal telepresence space has shown that the design of these systems would prioritize the Viewer's needs over the Streamer's needs [14]. To address this concern, we pursued a field study based approach in which pairs of participants would be able to interact with two interpersonal telepresence systems wherein each interpersonal telepresence system catered to one stakeholder of the pair dynamic. In the following sections, we discuss the remote streaming activities that were aimed to give participants perspective

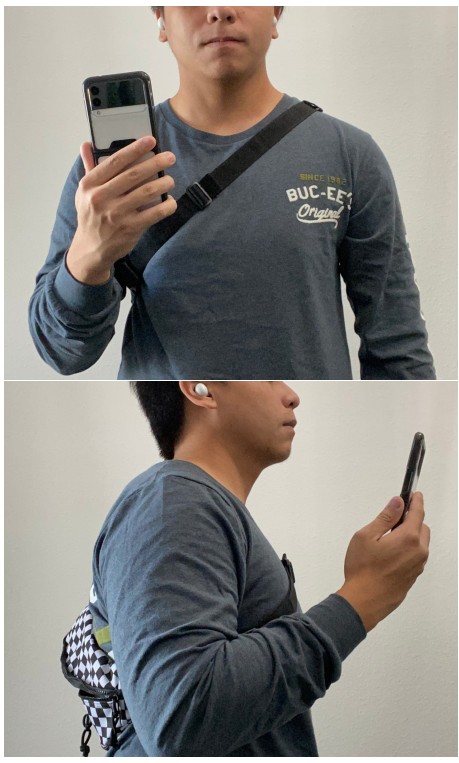

Figure 1: Streamer-Centric Design: Mobile device, adjustable low-profile backpack, and battery bank inside backpack

and real-world experience with interperonsal telepresence systems, and the conclusions and values they ultimately converged towards in regards to experience and features.

## 3.1 Participant Demographics

We distributed information about our study through our local university and online messaging forums in our city. Interested individuals were asked to fill out an online form to ensure eligibility and indicate date and time of the session they would be available for. Participants were required to be at least 18 years old, have a mobile device that supports calling and headphones, normal or corrected-to-normal vision, speak English, be able to walk for 40 minutes, and able to lift and carry 10 pounds. We also asked participants to provide their age and gender. Each pair of participants were required to know each other as friends, family members, or significant others prior to the session. Table 2 highlights our participant demographics in detail.

## 3.2 Apparatus

In the following sections we describe the apparatus utilized to carry out our Viewer/Streamer-centric remote streaming activities. In addition to basing our design choices on previous literature, we also utilized off-the-shelf technologies that would aid in user familiarity to the platforms and reduce possible steep learning curves from interacting with immersive technologies.

### 3.2.1 Streamer-centric apparatus

For the Streamer-centric condition, Streamer participants wore a low-profile backpack and were equipped with a mobile device to facilitate a *Zoom* [2] video-conferencing call for verbal and visual presence of the Viewer. The setup also included a power bank stored

---

[1] https://www.skype.com/en/

[2] https://zoom.us/

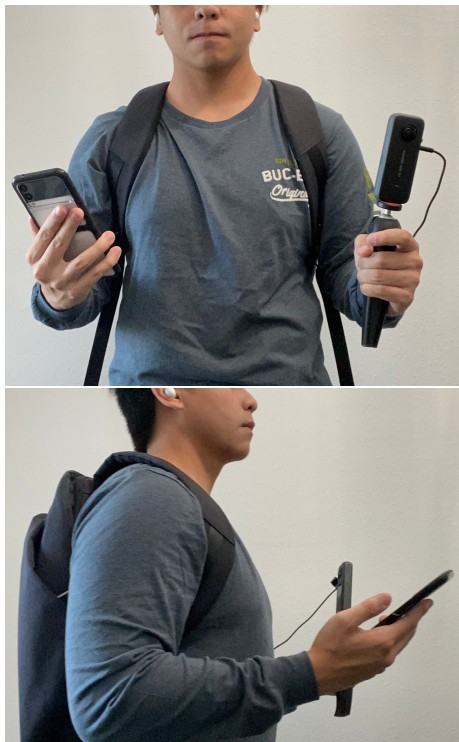

Figure 2: Viewer-Centric design: 360-degree camera, adjustable backpack, two (2) mobile devices (one in hand and one in backpack), and a battery bank inside backpack

in the backpack in order to charge the mobile device if needed during the remote streaming experience. These design choices were influenced by previous work that focused on utilizing ubiquitous devices, such as a cellular device, to facilitate a social telepresence experience [5, 6, 8, 16]. Figure 2 shows the setup the Streamer used in the Streamer-centric condition.

### 3.2.2 Viewer-centric apparatus

For the Viewer-centric condition, the Streamer was equipped with a backpack, two (2) mobile devices, a *Insta360 one X2* 360-degree camera[3], and a power bank stored in the backpack for charging purposes. Similar to the Streamer-centric condition, a *Zoom* video-conferencing call was used to facilitate verbal communication and visual presence. The addition of the 360-degree camera required the need of a private *YouTube*[4] live-stream as the platform. The purpose for choosing *Youtube* as our live-streaming platform is due to the ability to support a 360-degree video feed as well as allow the Viewer to pan around and interact with varying viewing angles as they pleased. The design choices for the Viewer-centric condition were influenced by previous work that investigated the collaborative capabilities between a Streamer and Viewer through a 360-degree camera medium [4, 18]. Figure 3 shows the setup the Streamer utilized to host the stream in the Viewer-centric condition. For both conditions, wireless earbuds facilitated two-way audio communication between the Streamer and the Viewer.

### 3.3 Study Procedure

Prior their session, we required pairs of participants to disclose a location of interest they would like to spend time together while using the interpersonal telepresence systems. The requirement for

---

[3]https://www.insta360.com/product/insta360-onex2
[4]https://www.youtube.com/

this location is that the location is either on our local university campus or within a 10-minute driving radius of our local university campus. We also required participants to choose their role in the Streamer and Viewer dynamic as the procedures differ between roles.

On the day of the session, the Streamer would meet with a researcher at the outlined location of interest, while the Viewer would meet another researcher in our lab. Each researcher provided a description of the study, a consent form they were required to sign, and instructions on how to use the Streamer-centric and Viewer-centric interpersonal telepresence setups. The participants were also informed that the entirety of the session will be both audio and video recorded and that all audio and video stored will be only viewed and accessible to the researchers and their research team.

During the remote streaming experience, the Streamer hosted both types of interpersonal telepresence setups. The order was randomized between groups. To facilitate a natural experience between the Streamer and Viewer, the researcher did not interfere or chaperone the Streamer throughout the remote streaming experience. The Streamer hosted the remote experience for 20 minutes per setup, for a total of 40 minutes of remote streaming.

For the Streamer-centric interpersonal telepresence setup, the Viewer viewed and communicated to the Streamer through a *Zoom* video-conference call. Audio communication was also hosted through the *Zoom* video-conference call. For the Viewer-centric interpersonal telepresence setup, the Viewer had the ability to view and pan around the Streamer's environment through a private *Youtube* 360-degree video stream and was able to view and communicate with the Streamer also through a *Zoom* video-conference call.

At the end of the remote streaming activity, the Streamer and Viewer were administered a semi-structured interview independently of one another and asked a series of questions related to their experiences with both interpersonal telepresence setups for approximately 30 minutes. Table 1 outlines the interview questions that participants were prompted based on their role in the interpersonal telepresence relationship. The questions were asked in the same order between each pair of participants. Participants were given a $15 Amazon gift card as compensation. All of the activities were audio and video recorded for analysis.

### 3.4 Data Analysis Approach

All sessions were audio and video recorded, and were transcribed by the authors with the assistance of Zoom. We conducted an inductive thematic analysis on both the remote streaming activity and semi-structured interview to better understand what our users truly valued when reflecting on the proposed setups [3]. We utilized open coding to log participants' explicit and implicit values from their utterances. We are interested in what features users truly want in an interpersonal telepresence system as well as what real world contexts are best suited for a more novel systems.

### 4 RESULTS

In this section we present our findings highlighted in Table 3 and provide further insight on our participants' experiences with the interpersonal telepresence setups

### 4.1 Participant Preferences

Overall, participants generally expressed that they liked using at least one of the systems, whether it be Viewer or Streamer. Participants especially expressed that the use of the 360-degree camera was unusual yet interesting in the Viewer-centric setup. Participants also have generally distinct situations where they would use each setup.

For the Streamer-centric setup, participants would most realistically use such a prototype for personal or exploratory use, e.g. if they were talking with friends or family or if they were in a new

Table 1: Interview Questions

| Streamers | Viewers | Both Roles |
|---|---|---|
| Describe how you felt while live-streaming on the 360-degree system and the video-conferencing system. | Describe how well you were able to see and hear the remote environment in both systems. | How well were you able to communicate with your partner? |
| | | What features of the video-conferencing system did you like and not like? |
| How well were you able to communicate with your partner? | Describe how socially present your partner felt in both systems? | What features should be changed to the video-conferencing system? |
| | | What features of the 360-degree system did you like and not like? |
| Why did you choose the role of Streamer? | Why did you choose the role of Viewer? | What features should be added, removed or changed to the 360-degree system? |
| | | If you were to choose one of the setups, which setup do you prefer and why? |
| | | What scenarios in your daily life would you consider using technology like this? |

place and would like to stream that new environment to people in remote locations. This is due to the immersive experience that the setup provides.

Although participants generally mentioned such a use for this prototype, another use that was mentioned was for security reasons, as the view provided would be able to show everything in the environment and not just one specific area, which would provide multiple perspectives.

*Like for security reasons that's also a good idea to have like one of these cameras set up somewhere.* - Streamer 6

For the Viewer-centric setup, participants would most realistically use such a prototype for everyday or professional use, e.g. if they were attending a meeting with a colleague or an online lecture, or doing some simple shopping. This is due to the simplicity and familiarity of the setup when using it.

*"Interacting on zoom in daily life is better, as compared to the 360 camera that what I would suggest in daily life, considering the daily life activities, and I would go with Zoom of course."* - Viewer 3

### 4.2 Participant Values

Participants held a wide range of values for both telepresence prototypes employed, which relate to both their own interests and the interests of the corresponding participant in the pair.

Viewers mainly valued interactive and independent viewing. This relates to the Viewer being able to immerse themselves as much as possible in the Streamer's environment in order to simulate the feeling that they are with the Streamer, along with being able to do as they please while using the prototype and not having to rely on the Streamer to do so.

*"So I would definitely prefer that 360 when me or my partner and any family member is going on a trip as a road trip or as sightseeing or just go for a ride probably and they can actually spin the camera around and what within my choice I could see here and there, without having to ask them to "hey can you flip the camera" yeah that's one convenience of 360 camera that conference meeting doesn't have."* - Viewer 6

Streamers mainly valued a minimalist telepresence prototype along with being able to interact with the physical environment while using the setup. This relates to the Streamer having to be mobile, and they wish to hold and wear as little as possible when doing so along with being able to both talk and see the Viewer. Along these lines, multiple improvements were recommended as changes to both systems to make them more minimalistic and user friendly.

*"Oh yeah making it more minimalistic."* - Streamer 3

Changes recommended for the Streamer-centric setup included attaching a wider lens angle for more Viewer immersion and having a mount for their smartphone. Changes to the Viewer-centric setup included having a single, multi-function device to serve as the central device of the setup to reduce the amount of equipment, reducing the backpack size, and having a mount for both the video conferencing smartphone and 360 degree camera.

*"Have an accessory that kind of like connects to the body in some way so that the weight is not just on the fingers, it could also be supported by like the rest or the entire arm, so that, so the user can hold it for extended periods of time."* - Streamer 10

Participants in general also held similar values such as wanting to be able to see each other during the use of either telepresence prototype, making sure that each other is safe, and wanting to directly interact with each other's surroundings.

*"When I went to see my friend, it was, like me, trying to get both of them to see and hear, but they couldn't hear each other, because of the headphones."* - Streamer 1

### 4.3 Social Activity

Social activity was present in multiple forms throughout the use of both telepresence prototypes. It consisted of interaction between the Streamer and Viewer, between the Streamer and their environment, and between the Streamer/Viewer and each other's environment.

We found that social activity between the Viewer and Streamer was desired, which is exactly what the prototypes were meant to facilitate in the form of telepresence, though it was definitely apparent that the Viewer-centric setup provided more social immersion for the Viewer while the Streamer-centric setup provided more connection and intimacy between both the Viewer and Streamer.

*"I think she was very socially present, I'd ask her questions, we'd have conversations, while I asked her like help me out with this choice should I get this or that and she'd answer back."* - Streamer 8

*"So for the 360 degree experience it's more of a fulfilled experience because not only do I get to hear the person so see the surrounding as if I was there, so, but for the second experience it's more um, I'd say it's more intimate but it's intimacy is not a very correct word because Facetime has become such a useful tool for everyone, and it has become a common tool for us to use, so I feel that I can readily use that tool, instead of a 360. 360 is more of an experience, and I would prefer that over to second."* - Viewer 6

*"Like with conferencing like I was able to see my partner as well, so I felt more connected than the 360 where she was able to see me,*

Table 2: Participant Demographics

| Session Number | Role | Gender | Age |
|---|---|---|---|
| One | Streamer | Female | 21 |
| | Viewer | Female | 21 |
| Two | Streamer | Male | 24 |
| | Viewer | Male | 23 |
| Three | Streamer | Male | 30 |
| | Viewer | Female | 27 |
| Four | Streamer | Female | 21 |
| | Viewer | Female | 21 |
| Five | Streamer | Male | 28 |
| | Viewer | Female | 29 |
| Six | Streamer | Male | 28 |
| | Viewer | Female | 24 |
| Seven | Streamer | Female | 21 |
| | Viewer | Female | 21 |
| Eight | Streamer | Female | 21 |
| | Viewer | Female | 21 |
| Nine | Streamer | Male | 19 |
| | Viewer | Male | 19 |
| Ten | Streamer | Male | 22 |
| | Viewer | Female | 26 |

*but I wasn't. I was able to see like on them live conferencing, I was also able to see what I'm recording."* - Streamer 7

In terms of social activity between the Streamer and their environment, we found that there were three social relationships they had with their environment; either they sought to actively interact with bystanders, they remained neutral and neither tried to interact or avoided possible social interactions, or they actively tried to avoid possible social interactions. This was influenced by how much they wished to achieve during the use of the prototypes and how much they communicated with their partner.

Finally, in some rare instances, either the Streamer or Viewer wished to interact with the other's environment directly without the other participant serving as a means to facilitate interaction. These relationships of social activity are accompanied by the location choices that Streamers and Viewers made. Participants mainly chose for the Streamer to go a place that was intimate or familiar to them, or to partake in an activity together.

*"That's kind of why I let her be Viewer, because I wanted to walk around and then I saw the arboretum, yeah yeah, and then I was like oh, this is a good chance to like walk around there, I think it was also if I bought myself a cookie and then I walked around like zones."* - Streamer 8

## 5 DISCUSSION

The results of our study provided insight into what confounding variables affect the Streamer and Viewer user experience in an interpersonal telepresence system. We also gained insight into the environmental contexts as well as the technological preferences that participants tended to converge towards for a future interpersonal telepresence system. The following section provides details on our findings.

### 5.1 Streamer social management variation (RQ1)

Through our study, we found that our Streamers' response to social situations and interactions varied greatly between each Streamer. Given that previous work typically alluded to the idea that social pressure or awkwardness may be an inhibiting factor to pursue real world experiences or interactions, our Streamer participants took one of three approaches to social experiences or bystander collocation: they Pursued social experiences, acknowledged bystanders, or avoided bystanders.

#### 5.1.1 Context affects Streamer approaches

Taking into the consideration of the three varying approaches our Streamers employed in regard to bystander or social interactions, we cannot definitely confirm that all Streamers will experience social pressures or discomfort when interacting with an interpersonal telepresence system.

Our work implies that one of the confounding factors that influence a Streamer's willingness to pursue social interactions is environmental context. For example, in regard to highly populated areas of our local university campus, our Streamers did not converge on a common approach, but instead chose approaches that best fit what they wanted to achieve through the remote experience. This also leads into the idea that our Streamers had very differing goals and intents throughout their interpersonal telepresence experience.

#### 5.1.2 Streamers are not uniform

We have highlighted the idea that context is a highly influencing factor in how Streamers interact with interpersonal telepresence systems. Continuing the idea regarding that Streamer's personal goals and intents may influence their insertion into social contexts, we consider that this may possibly lend itself to the personality type and preferences of the Streamer's themselves. What we have learned from this study is that we cannot generalize the attitudes and preferences a Streamer may have. Therefore, when designing an interpersonal telepresence system, we need to account for non-uniform Streamers and unique treatment of the system to cater to a given Streamer's preferences.

### 5.2 Novelty and Unique Experiences are linearly correlated (RQ2)

Across our pairs of participants, our two stakeholder groups held similar values with respect to the appropriate contexts where it is preferable or appropriate to utilize more unique and novel telepresence systems. We converge on two overarching contexts to generalize scenarios in which different telepresence systems may need to be employed: everyday experiences and unique experiences.

#### 5.2.1 System Familiarity is linked to Frequency

Everyday experiences are inclusive of events that a general user will encounter on a frequent basis. Such experiences, according to our participants, include casually connecting with friends/family/significant others, a one-on-one meeting in a professional context, and classroom environments. Participants also shared that they would typically prefer video-conferencing applications in general as they are *easier to use* and more *familiar* than more unique and novel systems. In cases where the remote experience is more focused on the human-to-human interaction and not inclusive of the environment, video-conferencing seemed to be ample enough for our participants to fulfill remote interactions and experiences with each other.

#### 5.2.2 Unique setups for Unique Experiences

On the opposite side of the spectrum, participants uniformly highlighted that in more infrequent, unique experiences (i.e. vacations, hiking trips, theme parks), a more interactive, novel

Table 3: Codebook Used For Qualitative Analysis

| Categories (N = 20) | Codes (N = 20) | Exemplar Quote |
|---|---|---|
| **Participants value more novel designs in more unique situations** | 360 camera better for exploratory/special occasion use (N = 20) | *"The 360 I could see that being used, maybe when you're like a museum exhibit and you don't want to walk through every painting, you could just walk straight through and then you know the Viewer can just slide left or right, whatever."* - Streamer 8, P15 |
| | Video conference is better for professional/everyday use (N = 20) | *"For [video-conferencing] I see it, more for like school meeting because it's like one on one you don't really need to see if there's other people around or whatnot."* - Viewer 7, P14 |
| **Participants valued a more minimalist and interactive interpersonal telepresence design** | Independent and Interactive Viewing (N = 9) | *"I could focus on enjoying what I'm doing and I have to worry about showing them the stuff. They can look around and do what they want to do and it's like they're actually there."* - Viewer 9, P18 |
| | Minimalist Streaming Setup Design (N = 9) | *"If I'm doing a livestream then [waist bag], because I just had like better access to the charger or the wires without having to take it off, with a backpack when you asked me to take out the phone and stuff I had to like take it off, but with [the waist bag] it was easy to access."* - Streamer 7, P13 |
| **Participants varied in terms of how much social interaction they desired** | Inserted themselves into social interactions (N = 3) | *"For example, I cross over by someone that was doing a TikTok dance, and I was able to to grab his Instagram account.""* - Streamer 2, P3 |
| | Did not care about social interactions (N = 4) | *"I didn't mind actually I wasn't actually aware of people around me that well, maybe because I was distracted like talking with her or something."* - Streamer 8, P15 |
| | Tried to minimize social interactions (N = 3) | *"I just didn't really want to get in the way of other people like that, they don't know what we're doing."* - Viewer 7; P13 |

system is more preferable to attempt to immerse an individual into a remote environment. Participants would further rationalize this concept by emphasizing the need for both the Streamer and Viewer should maintain a certain level of independence between one another. This would ultimately allow the Streamer and Viewer to interact with the remote environment of their own volition in an attempt to emulate having two individuals present in an environment instead of one. Despite this uniform consensus, we feel that this may be due in part to the sytems being rather novel. Pfeil et al. highlighted two important qualities regarding interpersonal telepresence systems: 1) more work is needed within this field, particularly in the form of logitudinal field studies, to learn of user values when novelty effects subside and 2) the concept of interpersonal telepresence systems is relatively new [14].

### 5.3 Bridging the gap in design based on Stakeholder commonalities (RQ3)

From our study, we were able to derive general parameters to designing interpersonal telepresence systems that, in our estimation, would be able to satisfy both the Viewer and Streamer. We consider two major themes: Interactivity and Minimalism.

#### 5.3.1 Stronger Presence through Interactivity and Immersion

Our Viewer participants tended to converge on the idea of being able to interact with the streamed video in a higher capacity than that of a typical video conference. When prompted about any improvements that could be made to the Streamer-centric design (Video-Conferencing application), Viewers hinted that if they were to improve the system, it would ultiamtely evolve into the Viewer-centric system in which the Viewer is able to pan around and view the environment for themselves. Furthermore, when prompted about any possible improvements they would consider to the Viewer-centric system, most Viewers suggested if it was at all possible to add additional controls or make the experience more immersive.

#### 5.3.2 Minimizing weight for real-world interaction

From the Streamer's perspective, Streamer participants converged on the idea that all the technology and equipment involved in the setup should be low profile. This is to resolve two main concerns the Streamers encountered during their experiences: comfort and flexibility. Though supported through previous literature, we found this conclusion intriguing in the sense that comfort and flexibility in a physical sense seemed to be a higher priority for our Streamer participants over comfort due to socially awkward experiences.

Comfort also seemed to be a recurring element as Streamers wanted to ensure that both the bags and equipment would be comfortable throughout their experiences. To achieve this, Streamers would often manipulate the given equipment in configurations that best suited them. For example, in our original figures, we showcased the Streamer-centric design with a one-strap backpack slung across the chest, however, we had participants that opted to wear the bag as a waist bag or over the shoulder similar a purse.

Flexibility was also another recurring theme across our paired sessions. Streamers wanted the ability to interact with the environment while providing their Viewer counterpart an enjoyable remote experience. However, they were at times not able to due to holding the equipment. Additionally, we informed Streamers they would be able to view their partner through the Zoom video-conference during the Viewer-centric portion of the sessions. Despite the suggestion, Streamers opted to rely on Zoom audio for communication and only carried the 360-degree camera to facilitate the remote experience for their partner. This heavily implies that Streamers want to be able to maintain an ability to interact with the physical environment without being impaired by equipment.

### 5.4 Converging towards an ideal telepresence system outlook

Based on our Streamer and Viewer participants collectively, it seems that the ideal interpersonal telepresence setup is comprised of a variety of factors. The overarching factor, however, is the context in which the interpersonal telepresence system is employed and whether or not the factor falls under a unique experience or an everyday experience.

Between the two contexts, there exists an overlap and consensus of what key values are important to users in an interpersonal telepresence system: the Context and Minimalism. The context plays an important role in determining whether or not it is a priority for a user to want to experience a remote environment. Furthermore, minimalism extends further than the physical and social comfort; it is also inclusive of how familiar the users are with the systems and whether or not they are willing to take on any learning loads

to better familiarize themselves with more unique systems. For these reasons, the factors that heavily influence the acceptance of an interpersonal telepresence system are the context in which they are employed and the technology remaining both physically and cognitively minimalist.

## 6 LIMITATIONS AND FUTURE WORK

Our results are not applicable to all general situations, contexts, and scenarios in which each system would be used. In our study, we asked our participants to choose a location on our local university campus or within a 10-minute drive of the campus. Many participants chose to stay within the campus in favor of familiarity. We cannot speak broadly about more diverse scenarios and contexts outside of hypothetical situations. To mitigate this bias towards familiarity, our aim in a future study is to broaden the locations in which participants are able to select from, which would provide the opportunity for the Streamer or Viewer to share a location they value or is new to.

Our study was less considerate to the Viewers as the Viewing experience was more or less the same with the only difference being able to view the environment through a 360-degree video stream or a video stream through Zoom. Participants desired the ability to further immerse themselves into the environment and even compared the Viewer-centric system to a video game. In a future study, more viewing/interaction options need to be available to the Viewer so that they are able to use varying interaction techniques with the systems.

Lastly, our study design provided limited exposure to each stakeholder-centric system. With 20 minutes to interact with each system, we ask if our participants experienced a novelty effect. Our work falls short of identifying the effects of long-term usage of each system. A future improvement would allow participants to interact with the systems for an extended period of time. This would allow for the systems to be utilized for a variety of purposes and periods of time, and would yield a more holistic understanding of participants' perceptions.

## 7 CONCLUSION

In this paper, we present our work towards identifying a balance of needs between Viewers and Streamers to apply towards a future interpersonal telepresence prototype system. Through a field study, we made use of prototypes based on prior literature and learned of the values and contexts Streamers and Viewers converged towards. Previous work has shown that telepresence designers have a tendency to create novel systems that augment the abilities of the Streamer but are rather obtrusive in a social contexts. By contrast, we found that our participants strongly favored a prototype that provided a balance of independence between the Streamer and Viewer. This prototype would allow the Streamer to freely interact with the physical environment in a minimalist fashion, while providing the Viewer with a highly interactive and entirely autonomous viewing experience of the remote environment. Through our work, we are moving towards the idea of integrating interpersonal telepresence systems within everyday life. With this goal in mind, our hope is to be able to strengthen interpersonal relationships between individuals and take the lessons learned from our experiences to apply towards more fulfilling and enriching interpersonal telepresence experiences.

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
