# OpenReview forum: "Context and Minimalism: User Evaluations on Two Interpersonal Telepresence Systems"
_graphicsinterface.org/Graphics_Interface/2022/Conference — Submitted to GI 2022_

### Official Review · Reviewer_R4rQ · 2022-04-04
**The prototype and study do not seem very well designed as to give new insights or provide much contribution to designers of these types of experiences.**

**Rating:** 3
**Confidence:** 5

**Review:**

The paper describes an experiment comparing a wearable Zoom apparatus to a Zoom + 360deg camera set up that enables a Viewer and a Streamer to experience what a teleconferencing scenario where one person is at the remote location (Streamer) while the other watches from a lab setting (Viewer). The main results indicated that the Streamer found the 360deg camera cumbersome, the Viewers wanted to be a be able to move the 360deg viewpoint around and that the Viewers liked having more of a view that the 360deg camera afforded. The authors conclude that their results imply that the context of the experience is an important design consideration as is targeting a minimal design.

The work continues the investigations in the literature of how to connect remote participants with a local Streamer and provides additional support to why ergonomic design is important for these systems and that novelty effects are important to consider when trying to make conclusions about how a technology will fare in a more general sense. While not novel, it provides valid lessons for other researchers.

The main concern I have is that the study seems quite superficial considering other work that is out there plus they essentially just tacked on 360deg video to see what people thought about it. Not surprisingly, given the makeshift approach used for making the prototype, feedback was received about the awkwardness/discomfort of carrying this particular prototype around making conclusions somewhat suspect about the other comments. So, ultimately, there’s not so much to be learned from the Streamers in this experiment design as ‘minimalism’ ends up as a euphemism for a poorly executed prototype. Likewise, the Viewers had quite limited use of the potential of the 360 deg video, so very little insight is to be gained there.

In the description of the two setups, it isn’t clear why one is particularly more Viewer centric than Streamer centric; for example, the amount of stuff the Streamer has to hold makes this not only Viewer centric, but also Streamer encumbered. It may have been reasonable to balance this by having the Streamer have the exact same apparatus in both conditions, for example; i.e. hold the 360deg camera, but have it turned off in the Streamer centric condition and turned on in the Viewer centric. Or, a Streamer centric interface would be the zoom + 360deg camera set up where they could control the views provided to the Viewer along with the 360deg camera attached in a way they wouldn’t need to hold it; like on a hat or something. Likewise, the Viewer centric would be more about how much control they have over the view space and/or the Streamer activities. These power/control relationship between a Viewer and Streamer was discussed in works such as the Cat in the Map [Anacleto et al, Interact 2015] work, for example. This work and other wearable literature where there are remote and local Viewer/Streamer would be appropriate to refer to.

The study didn’t seem to put effort to mitigate the novelty effect from the 360deg camera. So, some of the experiences and conclusions may be quite transitory in nature one people get used to the it. This also implies that it is more about control than what the person sees as that would be more long lasting issue as to how shared control between Streamer and Viewer is manages once a 360deg camera (or other views) are available.

There are a few typos:
5.3.1 typo: ultiamtely -> ultimately
5.3.2 type: similar to purse -> similar to a purse

In summary, the prototype and study do not seem very well designed as to give new insights or provide much contribution to designers of these types of experiences.

---

### Official Review · Reviewer_afmW · 2022-04-13
**Recommend revision. Encourage more literature review and draw from other communities use of videophone. Positive aspect was the 360 camera, expand more on this.**

**Rating:** 6
**Confidence:** 5

**Review:**

The first two paragraphs in the introduction don't flow well. The first paragraph doesn't connect with rest of the paper, eg. general statement about health effects or "confined to living space". I recommend language/word choices to be more objective, eg in first paragraph of introduction: "Now either confined to their living space". the word confined has negative nuance. Stating importance of the remote collaboration in first paragraph with no evidence or data to support this makes it weak. It is recommended to cite data that shows importance of the collaboration.

The usage of the word "novelty" in third paragraph of introduction is confusing. Where and how is it new? I recommend this concept be defined in introduction to connect with discussion section 5.2.

Telepresence has been around as early as 1970's, Rochester Institute of Technology had technological devices.  There has been research on experiences with telepresence technology, called videophones, used by Deaf community. I recommend that authors show how their research builds on to previous research in this area. More literature review is needed in the introduction.

There is a bit of disconnect between introduction focusing on  and the study in which participant walks around with the streamer prototype.

The trade-offs need to be more explicit.

I also recommend defining social activity. The term was used in the paper and was one of the values measured. However the term was not defined. The term is very broad in scope. The technological variables were set out, but social interactions were not defined clearly.

Testing out the 360 degree camera as novel experience as part of one-to-one interaction was part of this paper is I found useful and engaging to read.  I encourage pursuing this more and adding more options for sharing comments one-on-one interaction using the 360 degree paper to work across more range of abilities/and disabled people.

I don't think Demographics table 2 is needed, a summary is sufficient.

I encourage you to also look at literature in which people carry around devices, such as how long can they carry it around? Or is it dependent on their physical ability to hold something of a certain weight and duration of time? Eg. people hold smart devices for face-time. What is the trade-off for holding devices and quality eg tired hands, moving/walking making video bounce?

---

### Official Review · Reviewer_nwm2 · 2022-04-14
**A study that explicitly compares handheld telepresence system user preferences for both the remote and local viewer's needs.**

**Rating:** 7
**Confidence:** 4

**Review:**

The authors conduct an exploratory comparison of two telepresence system, each trying to cater to either the remote or telepresent viewer. The key difference was a 360 degree camera that allowed a viewer to see and pan around in the environment. They found different preferences depending on role: the remote person preferred minimal equipment that was comfortable, while the viewer's preference depended on context, but new environments should use the 360 camera, with more intimate social encounters being served fine with existing telepresence technologies.

Pros:
- a nice comparison and a neat idea to focus also on both sides of the telepresence equation.
- clear and easy to understand results
- well written paper

Cons:
- weak, obvious results
- somewhat ignores the effort in other spaces like AR, VR, and HRI

Overall I think this is a good paper, and I like the focus, experiment, methodology, and findings. It is unfortunate that the results come down to "people prefer technology designed for them" and "context matters". Extrapolating from common UX and HCI design findings over the decades, this is not groundbreaking research.  However, it is useful research, as mentioned there are many benefits. Chief of which is further focus on considering both sides of telepresence, and explicitly comparing systems to untangle the tensions between the two in design. I think that core idea and the method is useful.

The other criticism is a lack of discussion of the wide range of telepresence work in multiple fields. A few examples are given in VR, HRI, and AR, but really the work in telepresence goes deep there, especially acknowledging and focusing on the social impact of those technologies.  Just some selections from my own HRI telepresence library:

1. Irene Rae and Carman Neustaedter. 2017. Robotic telepresence at scale. Conference on Human Factors in Computing Systems: 313–324.
2. Carman Neustaedter, Gina Venolia, Jason Procyk, and Daniel Hawkins. 2016. To beam or not to beam: A study of remote telepresence attendance at an academic conference. Proceedings of the ACM Conference on Computer Supported Cooperative Work, CSCW 27: 418–431.
3. Matthew Rueben, Frank J Bernieri, Cindy M Grimm, and William D Smart. 2017. Framing Effects on Privacy Concerns about a Home Telepresence Robot. In Human-Robot Interaction, 435–444.
4. Erina Okamura and Fumihide Tanaka. 2016. A Pilot Study About Remote Teaching by Elderly People to Children Over a Two-way Telepresence Robot System. In 2016 11th ACM/IEEE International Conference on Human-Robot Interaction (HRI), 489–490.
5. Katherine M. Tsui, Adam Norton, Daniel J. Brooks, Eric McCann, Mikhail S. Medvedev, and Holly A. Yanco. 2013. Design and development of two generations of semi-autonomous social telepresence robots. In 2013 IEEE Conference on Technologies for Practical Robot Applications (TePRA), 1–6.
6. J. Wentzel, D.J. Rea, J.E. Young, and E. Sharlin. 2015. Shared presence and collaboration using a co-located humanoid robot. In HAI 2015 - Proceedings of the 3rd International Conference on Human-Agent Interaction.
7. Annica Kristoffersson, Silvia Coradeschi, and Amy Loutfi. 2013. A review of mobile robotic telepresence. Advances in Human-Computer Interaction 2013: 1–17.
8. Kazuaki Tanaka, Naomi Yamashita, Hideyuki Nakanishi, and Hiroshi Ishiguro. 2016. Teleoperated or autonomous?: How to produce a robot operator’s pseudo presence in HRI. In Human-Robot Interaction, 133–140.

The authors don't need to cite any of all of these, I just feel an expanded related work section on XR and HRI for social telepresence would be useful for readers, and help better situate this work in the literature.

So, that stated, I do not think the related work is a fatal flaw, but could be updated for a camera ready easily. I also think the weak results are okay, as there are still results, and even if it is extensions of known HCI results, I believe it's useful to confirm these in a new setting, before further moving on (essentially, checking assumptions for the field). There are also questions and interesting data points that make me want to do further research in the area, and I think that's an important part of any paper. Additionally, it's mostly well written and easy to read (though I think it could be made much more concise). Thus, I will recommend weak acceptance.


small note: When giving exemplary quotes to indicate the opinion of multiple participants, I think it is useful to state how many participants shared opinion, E.g., "4/15 participants thought something like: <quote>". Like in Table 3, but put this in the text as well.

---

### Decision · Program_Chairs · 2022-04-17

Reject